# Effect of Co-Doping on the Magnetic Ground State of the Heavy-Fermion System CeCu₂Ge₂ Studied by Neutron Diffraction

**Rajesh Tripathi** [1,2,*] , **Dmitry Khalyavin** [1] , **Shivani Sharma** [1] , **Devashibhai Thakarshibhai Adroja** [1,3,*]
and **Zakir Hossain** [2,*]

1   ISIS Facility, Rutherford Appleton Laboratory, Chilton, Didcot OX11 0QX, Oxon, UK;
    dmitry.khalyavin@stfc.ac.uk (D.K.); phy.shivanisharma@gmail.com (S.S.)
2   Department of Physics, Indian Institute of Technology, Kanpur 208016, India
3   Highly Correlated Matter Research Group, Physics Department, University of Johannesburg, P.O. Box 524,
    Auckland Park 2006, South Africa
*   Correspondence: raj7tpi@gmail.com (R.T.); devashibhai.adroja@stfc.ac.uk (D.T.A.); zakir@iitk.ac.in (Z.H.)

**Abstract:** The antiferromagnetic phase transition of the heavy-fermion system $Ce(Cu_{1-x}Co_x)_2Ge_2$ for $x = 0.05$ and 0.2, showing up in specific heat, magnetic susceptibility, and muon spin relaxation ($\mu$SR) data, has been further investigated. The neutron diffraction (ND) results show that Co-doping drastically reduces the moment size of Ce, without a qualitative change in the magnetic structure of the undoped compound $CeCu_2Ge_2$. An incommensurate magnetic propagation vector $k = (0.2852, 0.2852, 0.4495)$ with a cycloidal magnetic structure with a Ce moment of 0.55 $\mu_B$ in the $ab$-plane has been observed for $x = 0.05$. Although for $x = 0.2$ the specific heat and magnetic susceptibility data reflect a phase transition with a broad peak and the muon relaxation rate shows a sharp peak at $T = 0.9$ K, our ND data dismiss the possibility of a long-range magnetic ordering down to 50 mK. The ND data, along with previously reported results for $x = 0.2$, are interpreted in terms of the reduced ordered state magnetic moments of the $Ce^{3+}$ ion by Kondo screening and the presence of dynamical short-range magnetic correlations.

**Keywords:** heavy-fermion system; antiferromagnetism; neutron diffraction

## 1. Introduction

Over the past few decades, heavy-fermion systems containing a sub-lattice of $4f$ or $5f$ elements have been intensively studied [1–10]. Competing interactions often reveal different ground-state properties in these compounds. The inter-site Ruderman–Kittel–Kasuya–Yosida (RKKY) interaction favours a long-range magnetic order and the onsite Kondo effect suppresses magnetic ordering by screening the local magnetic moments. Tuning the relative strength of the onsite Kondo and intersite RKKY exchange interactions leads to various phenomena, such as long-range magnetic order, quantum critical fluctuations, unconventional superconductivity, and heavy-Fermi- and non-Fermi-liquid (NFL) behaviors [2–10]. In particular, the heavy-fermion compound $CeCu_2Ge_2$ is an ideal system to study as its anti-ferromagnetic (AFM) ground state can be easily tuned by a magnetic field or an external or chemical pressure [11–13]. $CeCu_2Ge_2$ is a magnetically ordered Kondo lattice with an AFM phase transition $T_N = 4.1$ K and a Kondo temperature $T_K = 6$ K [14]. A sinusoidal spin-density-wave-type incommensurate magnetic ordering with a propagation vector of $k = (0.283, 0.283, 0.538)$ at $T = 40$ mK has been reported [15]. Over the past few years, more detailed studies of the electronic properties and the magnetic structure under chemical or external pressure have been reported on $CeCu_2Ge_2$ using neutron diffraction (ND) experiments [15–17].

Recently, we reported the tetragonal Kondo lattice series $Ce(Cu_{1-x}Co_x)_2Ge_2$ using bulk and muon spin relaxation ($\mu$SR) measurements [18,19]. The cobalt substitution compresses the unit cell volume and increases the hybridization between the $4f$ electrons and

the conduction band states. For intermediate concentrations, two distinct AFM phase transitions were anticipated. The first type of transition, the so-called local moment type of AFM, persists up to $x = 0.1$. For higher $x$ ($0.1 < x < 0.6$), the development of a heavy-fermion band magnetism has been predicted, and finally the system changes its state to a heavy Fermi liquid close to $x = 1$. Moreover, NFL behavior develops as $T_N \to 0$ K for $x \sim 0.6$. This feature was characterized by the formation of magnetic clusters in a non-magnetic background known as the Griffiths phase [18,19]. The AFM phase in the second regime is associated with pronounced anomalies in the temperature dependence of the magnetic susceptibility but a weaker anomaly in the specific heat. In addition, the $\mu$SR experiment also confirms that the samples with $x = 0$ and 0.2 have magnetic ordering below $T = 4$ and 0.8 K, respectively, but the true nature of the magnetic phase could not be established yet. To shed more light on the unusual magnetic properties and the complex magnetic phase diagram of Ce(Cu$_{1-x}$Co$_x$)$_2$Ge$_2$, we performed neutron powder diffraction measurements for $x = 0.05$ and 0.2 samples. We anticipate that the ND experiment will give a difference in magnetic structure for Ce(Cu$_{1-x}$Co$_x$)$_2$Ge$_2$ between $0 \leq x \leq 0.1$ and $0.1 < x < 0.6$ samples, similar to the Ni-doped CeCu$_2$Ge$_2$ system, because the substitution of Ni at the Cu site also compresses the unit cell and increases the hybridization [20]. Compared to Ni, small Co concentrations strongly suppress the magnetic ordering temperature because, in addition to the volume effect, the change in the electronic structure plays an important role.

## 2. Experimental Methods

Polycrystalline samples of Ce(Cu$_{1-x}$Co$_x$)$_2$Ge$_2$ for $x = 0.05$ and 0.2 were prepared by arc melting stoichiometric amounts of high-purity elements in an argon atmosphere. Initial characterization of the samples was performed by X-ray diffraction with Cu-K$_\alpha$ radiation at room temperature, magnetization, and transport measurements. Neutron powder diffraction experiments were performed using the time-of-flight WISH diffractometers at the ISIS Pulsed Neutron and Muon Source, Rutherford Appleton Laboratory, United Kingdom [21]. The powdered Ce(Cu$_{1-x}$Co$_x$)$_2$Ge$_2$ samples were lightly packed in a thin-walled copper can (diameter 3 mm). The low temperature was achieved by cooling the sample inside a He-3 cryostat (for $x = 0.05$ sample) and a dilution fridge (for $x = 0.2$ sample) using He-exchange gas inside the In-sealed Cu can to ensure good thermal contact at low temperatures. In order to determine the magnetic structure, ND data for $x = 0.05$ and 0.2 samples were collected down to 0.28 K and 50 mK, respectively, with long counting (5 h per run). We also collected the diffraction data at several temperatures between 0.28 and 5 K with a shorter counting time (30 min per point) for $x = 0.05$ to investigate the temperature dependence of the order parameter as well as any change in the magnetic structure with temperature. Rietveld refinement technique was used to refine the low-temperature nuclear pattern using Fullprof program [22].

## 3. Results and Discussion

A neutron powder diffraction pattern of Ce(Cu$_{1-x}$Co$_x$)$_2$Ge$_2$ for $x = 0.05$ at temperatures above the magnetic ordering ($T = 2$ K) is shown in Figure 1a. The Fullprof structural refinement of this pattern reveals a single-phase ThCr$_2$Si$_2$-type (space group $I4/mmm$) tetragonal structure. Along with the main phase, copper (because we used a Cu can) with space group $Fm-3m$ [23] was also detected. The lattice parameters of the nuclear reflections were determined to be 5 K and 0.28 K and are listed in Table 1. The ND data thus confirm that the crystal structure for $x = 0.05$ down to 0.28 K remains the same as that at room temperature.

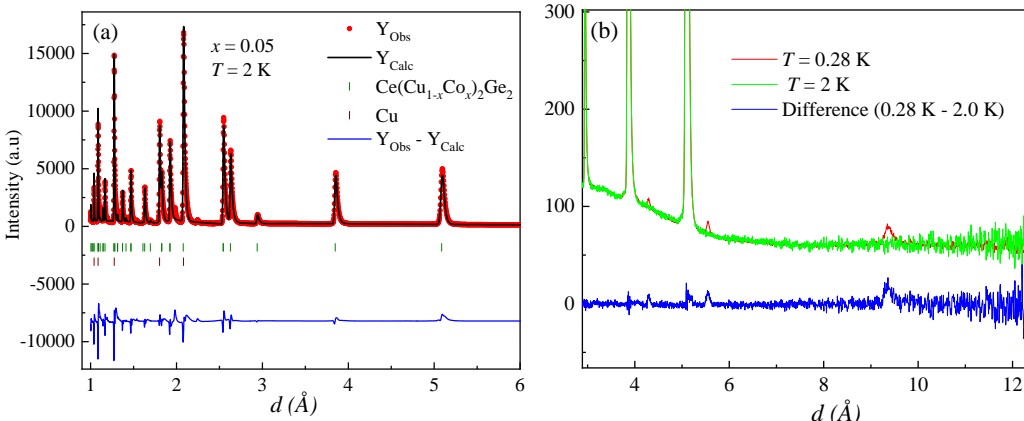

**Figure 1.** (**a**) Neutron powder diffraction patterns of $Ce(Cu_{0.95}Co_{0.05})_2Ge_2$ from one of the detector banks of the WISH diffractometer at $T = 2$ K. The solid line represents the Rietveld refinement profile fit for the $ThCr_2Si_2$-type body-centered tetragonal (space group $I4/mmm$) structure. A small amount of impurities of copper ($Fm-3m$), from the Cu-sample can, is included in the refinement. (**b**) The expanded view shows the differences between 2 K (green) and 0.28 K (red) data, where the magnetic Brag peaks are quite visible. The blue curve at the bottom shows the temperature difference data 0.28–2 K.

**Table 1.** Crystallographic parameters obtained from Rietveld refinement of powder XRD (room temperature) and ND data of $Ce(Cu_{1-x}Co_x)_2Ge_2$ ($x = 0.05, 0.2$) at different temperatures with the body-centered tetragonal $ThCr_2Si_2$-type structure (space group $I4/mmm$).

| | $x = 0.05$ | | | $x = 0.2$ | | |
|---|---|---|---|---|---|---|
| | **XRD** | **ND** | **ND** | **XRD** | **ND** | **ND** |
| | **(300 K)** | **(5 K)** | **(0.28 K)** | **(300 K)** | **(0.9 K)** | **(50 mK)** |
| Lattice parameter | | | | | | |
| $a(\text{Å})$ | 4.1759(1) | 4.1689(5) | 4.1680(1) | 4.1744(4) | 4.1729(5) | 4.1708(2) |
| $c(\text{Å})$ | 10.1860(7) | 10.1702(3) | 10.1711(1) | 10.1225(1) | 10.1230(8) | 10.1149(2) |
| $V(\text{Å}^3)$ | 177.6249 | 176.7518 | 176.6930 | 176.3908 | 175.2727 | 175.9545 |
| Atomic coordinate | | | | | | |
| $z_{Ge}$ | 0.3767 | 0.3776 | 0.3777 | 0.3753 | 0.3766 | 0.3766 |
| Refinement quality | | | | | | |
| $\chi^2$ | 1.67 | 19.4 | 20.3 | 2.13 | 17.4 | 19.3 |
| $R_P(\%)$ | 16.5 | 17.93 | 15.01 | 19.6 | 17.93 | 18.82 |
| $R_{WP}(\%)$ | 20.4 | 15.32 | 14.91 | 26.3 | 19.72 | 21.11 |

A comparison of the diffraction pattern at 2 K and 0.28 K in an expanded scale reveals the appearance of additional weak magnetic Bragg reflections below $T = 2$ K for $x = 0.05$ (Figure 1b). The reflections exhibit a critical behavior (Figure 2), and a power-law fitting of the integrated intensity plotted as a function of temperature yields the transition temperature $T_N = 2.0(2)$ K and the critical exponent $\beta = 0.31(2)$. $T_N$ is in excellent agreement with the results of $\mu$SR, magnetization, and heat capacity measurements [19]. The magnetic peaks can be indexed with the incommensurate propagation vector $k = (0.2852, 0.2852, 0.4495)$, which is close to the propagation vector reported by Singh et al. [15] for the undoped $CeCu_2Ge_2$ compound. The quantitative refinement of the magnetic intensities was therefore approached based on the magnetic ground state of $CeCu_2Ge_2$. The model provided a reasonably good refinement quality (Figure 3) but with a significantly reduced moment size 0.55(1) $\mu_B$ in comparison with the undoped counterpart (1.04(4) $\mu_B$). The model implies a gradual rotation of the Ce magnetic moments, confined within the

tetragonal *ab*-plane, upon propagation through the crystal (Figure 4). Apparently, the doping significantly increases the Kondo screening, as further evidenced by the lack of any detectable magnetic signal in the $x = 0.2$ sample (Figure 5a).

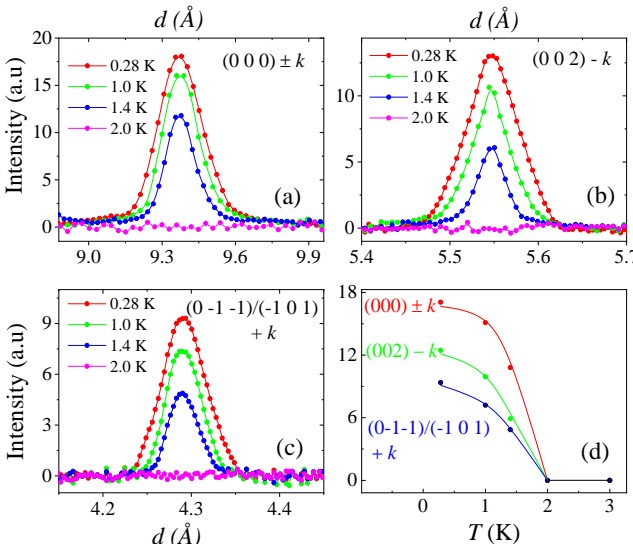

**Figure 2.** Magnetic Bragg peaks (**a**) (0 0 0) $\pm k$, (**b**) (0 0 2) $- k$, and (**c**) (0 $-1$ 1)/($-1$ 0 1) $+ k$ at various temperatures, and (**d**) the integrated intensity of the peaks (0 0 0) $\pm k$, (0 0 2) $- k$, and (0 $-1$ 1)/($-1$ 0 1) $+ k$ versus temperature for $x = 0.05$. The solid lines are fit to the data with power-law behavior.

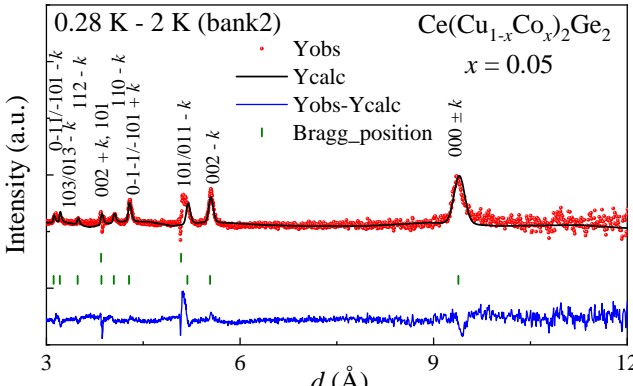

**Figure 3.** Rietveld refinement of the magnetic intensities obtained by subtraction of high-temperature (2 K) paramagnetic data from the low-temperature (0.28 K) data of Ce(Cu$_{1-x}$Co$_x$)$_2$Ge$_2$ with $x = 0.05$. The solid black lines show the fit. The difference between the experimental and calculated intensities is shown by the blue curves at the bottom. The olive vertical ticks show the position of the magnetic Bragg peaks (bottom) and the structural Bragg peaks (top).

The Rietveld refinement confirms that the $x = 0.2$ sample also crystallizes in the tetragonal ThCr$_2$Si$_2$-type structure space group $I4/mmm$. The refined lattice parameters, atomic position parameters, and thermal parameters are given in Table 1. As we noted a magnetic anomaly in $\chi(T)$, $C(T)$, and in the $\mu$SR measurements at around $T = 0.8$ K for $x = 0.2$, we can expect additional magnetic reflections in the difference ND data for above and below the ordering temperature, i.e., $T_N = 0.8$ K. Comparing the data collected at two temperatures (Figure 5b), i.e., at $T = 0.9$ K and 50 mK, we do not observe magnetic signals. This is because the moment is too small (apparently below the detectable limit). This is not unexpected: if $x = 0.05$ of Co reduced the moment from 1 to 0.5 $\mu_B$, it is quite natural that the moment size in $x = 0.2$ is outside the resolution limit. Fitting of the difference ND data in the model used to refine the magnetic structure of the $x = 0.05$ sample did not

yield a statistically significant moment size with the three-sigma interval around 0.2 $\mu_B$, which was taken as the top limit of the ordered moment for the $x = 0.2$ sample. This is in agreement with the observation of the dynamical electronic relaxation down to 0.35 K seen in the $\mu$SR data as discussed in our previous report [19]. Our $\mu$SR data neither showed any sign of frequency oscillations, which one expects for a long-range magnetic ground state with small moments ordering, nor a loss of 2/3 initial asymmetry, which one expects for a larger moment long-range magnetic ordered ground state. It is to be noted that in $CeCu_2(Si_{1-x}Ge_x)_2$, for $x \leq 0.4$, no magnetic intensities could be detected either by powder or by single-crystal ND [24]. This also supports the Kondo screening of the Ce moments for $x \leq 0.4$.

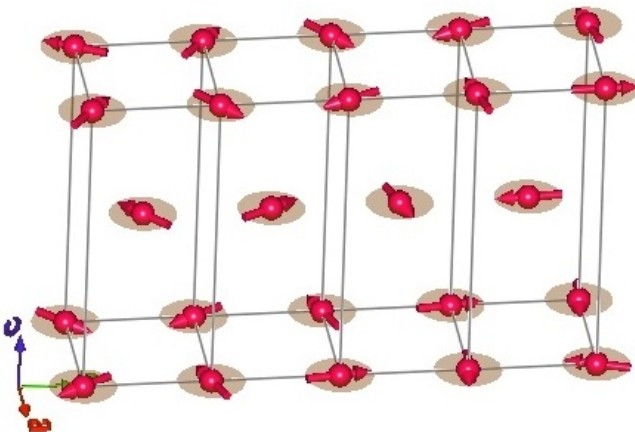

**Figure 4.** The incommensurate magnetic structure of $Ce(Cu_{1-x}Co_x)_2Ge_2$ ($x = 0.05$) with magnetic propagation vector $k = (0.2852, 0.2852, 0.4495)$ obtained from the refinement of ND pattern at 0.28–2 K. The solid lines show the unit cell, and the magnetic moments are shown by red arrows at the Ce atom position presented by red balls.

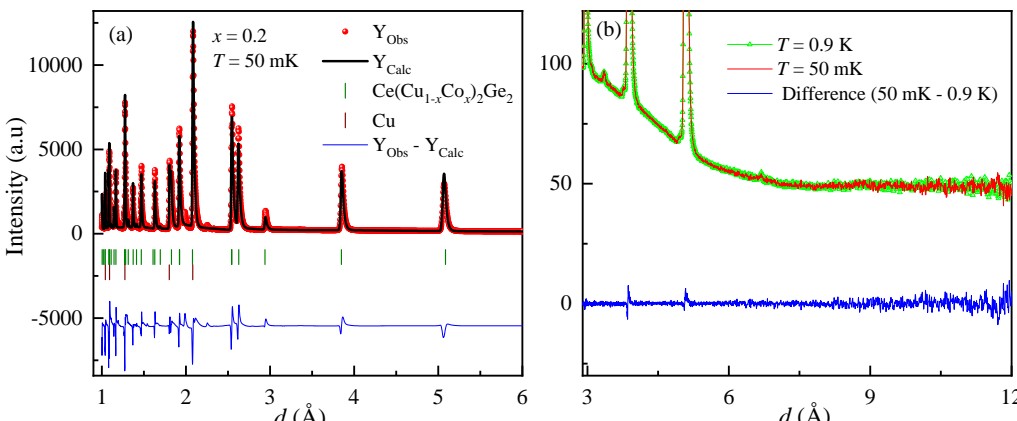

**Figure 5.** (**a**) Neutron powder diffraction patterns of $Ce(Cu_{0.8}Co_{0.2})_2Ge_2$ at $T = 50$ mK. The solid line represents the Rietveld refinement profile fit for the $ThCr_2Si_2$-type body-centered tetragonal (space group $I4/mmm$) structure. The refinement includes a small amount of copper impurities ($Fm-3m$). (**b**) No magnetic Brag peak could be seen in the expanded view of the difference between 0.9 K and 50 mK data.

## 4. Conclusions

We have investigated the magnetic structure of $Ce(Cu_{1-x}Co_x)_2Ge_2$ using neutron powder diffraction for $x = 0.05$ and 0.2 in the temperature range $0.28 \leq T \leq 5$ K and $0.05 \leq T \leq 5$ K, respectively. The results of ND reveal that Co-doping significantly reduces the moment size of Ce while leaving the magnetic structure of the undoped compound $CeCu_2Ge_2$ almost unchanged. For $x = 0.05$, we observed magnetic reflections with incom-

mensurate magnetic propagation vector $k = (0.2852, 0.2852, 0.4495)$, the magnetic structure obtained from the refinement corresponds to a cycloidal structure, and the Ce moments are antiferromagnetically coupled in the *ab* plane. The value of the magnetic moment of Ce at 0.28 K is 0.55(1) $\mu_B$/Ce-atom. This value is slightly smaller than the ordered state moment of $CeCu_2Ge_2$, i.e., 1.04(4) $\mu_B$, which could be due to the presence of the Kondo effect. This magnetic structure is identical to that of $CeCu_2Ge_2$ with propagation vector $k = (0.28, 0.28, 0.54)$. However, for $x = 0.2$, no magnetic Brag peak could be detected, which implies either complex short-range dynamical magnetic fluctuations or a further reduction in the ordered state Ce magnetic moments for Co concentrations $x \geq 0.1$, which are below the detectable limit. The results of the present study will be important in future investigations of the low-energy magnetic excitations in these materials.

**Author Contributions:** Conceptualization, Z.H. and R.T.; methodology, R.T., D.K. and D.T.A.; validation, R.T., D.T.A. and D.K.; formal analysis, R.T., D.K., S.S. and D.T.A.; investigation, R.T.; resources, Z.H.; data curation, D.K. and D.T.A.; writing—original draft preparation, R.T. and D.T.A.; writing—review and editing, R.T., D.T.A., D.K., S.S. and Z.H.; visualization, R.T. and D.T.A.; supervision, Z.H. and D.T.A; project administration, Z.H.; funding acquisition, Z.H. All authors have read and agreed to the published version of the manuscript.

**Funding:** This research was supported by the Department of Science and Technology, India (SR/NM/Z-07/2015) for the financial support and Jawaharlal Nehru Centre for Advanced Scientific Research (JNCASR) for managing the project. Funding was also supported by the Royal Society of London and EPSRC-UK (Grant No. EP/W00562X/1). R.T. thanks the Indian Nano-mission for a post-doctoral fellowship.

**Institutional Review Board Statement:** Not applicable.

**Informed Consent Statement:** Not applicable.

**Data Availability Statement:** Data will be made available on request.

**Acknowledgments:** We gratefully acknowledge the ISIS facility for the beam time on WISH(RB1868007). R.T. and D.T.A. would like to thank Pascal Manuel of the ISIS Facility UK and Premakumar Yanda of the JNCASR, India, for the helpful discussion.

**Conflicts of Interest:** The authors declare that they do not have any known conflicts of interest.

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
