# Peer review of "Effect of Co-Doping on the Magnetic Ground State of the Heavy-Fermion System CeCu2Ge2 Studied by Neutron Diffraction"

_magnetochemistry, doi:10.3390/magnetochemistry9050115_

Round 1

Reviewer 1 Report

About x = 0.2 at 50 mK, magnetic peak is not observed (Fig. 5 (b)) whereas a magnetic anomaly is observed in χ(T), C(T), and in μSR measurements.  Authors insisted the moment is too small (< 0.2 μB) to be detected.  But in Fig. 1 (b), where x = 0.05 at 0.28 K, the magnetic peak with magnetic moment as small as 0.55(1)μB is detected.  Since it is related with the conclusion, the referee c to discuss the limit of the detectable magnetic moment in viewpoint of the error propagation of the current data.  

Author Response

Please see attcahed file

Reviewer 2 Report

Over all the paper is well written and the discussions are clear.

The Rietveld refinement could be improved. The quality factors for the neutron data are all rather large. Furthermore in figures 1 and 5 the difference curves show the traditional shape corresponding to a discrepancy between the observed and calculated peak positions. 

On the last full sentence on the 1st page, I was left wondering what they reported.  I think it needs to say ,"... we reported bulk measurements on the tetragonal ..."

Both should be addressed before publication.

Reviewer 4 Report

This manuscript describes the effect of cobalt substitution on the long range ordered phase of CeCu2Ge2.  It also makes comparisons to prior studies of Ni substitution of the same compound.  The article is well written and makes a good contribution to the literature.  I do have minor issues that should be addressed before i can fully endorse this manuscript for publication.

1) first sentence, "during the past few decades,......" but only one reference is given.  please include references from multiple decades of study of the topic to make it clear that these materials were "intensively studied".

2) in first paragraph of experimental methods... I am concerned with thermalization of the sample temperature.  please comment if sample is a metal or an insulator.

3) Figure 2. please describe the curves for panel (d) in the caption of the figure.

4) diffraction runs were for 5 hours.  How much time was spent to thermalize the sample before beginning the run?  Are you able to look at the data in the first hour and compare it to the 5th hour to verify that it was thermalized?  this may not be possible, but one should be able to know the time spent to thermalize the sample before data collection.

5) error bars of refined parameters should be added to table 1.

6) for the refinement of the order parameter, was the value of Tn=2K fixed?  or was it allowed to vary?  if it varied, can you please list the errorbar of the refined value?

7) This is the most important comment.  can you please also show the muSR data?  THe thermodynamic data would be interesting to see also, but the muSR data would complement the neutron diffraction data and make the discussion more complete and compelling.   

Author Response

Please see attcahed file
